# Population attributable fractions of cancer mortality related to indoor air pollution, animal contact, and water source as environmental risk factors: Findings from the Golestan Cohort Study

**Negar Rezaei[1,2‡], Maryam Sharafkhah[1‡], Yalda Farahmand[3], Sadaf G. Sepanlou[1], Sahar Dalvand[1], Hossein Poustchi[2,4], Alireza Sajadi[2,5], Sahar Masoudi[1], Gholamreza Roshandel[6], Masoud Khoshnia[6], Layli Eslami[2], Mahboube Akhlaghi[1], Alireza Delavari[2,5]***

1 Digestive Disease Research Center (DDRC), Digestive Disease Research Institute, Tehran University of Medical Sciences, Tehran, Iran, 2 Digestive Diseases Research Institute, Tehran University of Medical Sciences, Tehran, Iran, 3 School of Medicine, Terhan University of Medical Sciences, Tehran, Iran, 4 Liver and Pancreatobiliary Disease Research Center, Digestive Disease Research Institute, Tehran University of Medical Sciences, Tehran, Iran, 5 Digestive Oncology Research Center, Digestive Disease Research Institute, Tehran University of Medical Sciences, Tehran, Iran, 6 Golestan Research Center of Gastroenterology and Hepatology, Golestan University of Medical Sciences, Gorgan, Iran

‡ NR and MS Equal First Author
* alirezadelavari91@gmail.com, delavari@tums.ac.ir

## Abstract

### Background

Environmental risk factors are significant contributors to cancer mortality, which are neglected.

### Purpose

This study aimed to estimate the population attributable fraction of cancer mortality due to the environmental risk factors.

### Methods

Golestan cohort study is a population-base cohort on 50045 participants between 40–75 with about 18 years of follow up. We detected 2,196 cancer mortality and applied a multiple Cox model to compute the hazard ratio of environmental risk factor on all cancer and cancer-specific mortality. The population attributable fraction was calculated, accordingly.

### Results

Biomass fuels for cooking, as an indoor air pollution, increased the risk of colorectal, esophageal, gastric cancer, and all-cancer mortality by 84%, 66%, 37%, and 17% respectively. Using gas for cooking, particularly in rural areas, could save 6% [Population Attributable

**Data Availability Statement:** Data described in the manuscript, data dictionary, and analytic codes will be made available upon request (pending filling an application to access the database of the Golestan Cohort Study (https://dceg2.cancer.gov/gemshare), review and approval by the study principal investigators).

**Funding:** The Golestan Cohort Study work was funded by the Tehran University of Medical Sciences (grant number: 81/15), Cancer Research UK (grant number: C20/A5860), the Intramural Research Program of the U.S. National Cancer Institute, National Institutes of Health, and the International Agency for Research on Cancer. The presented analysis was supported by Digestive Disease Research Institute, Tehran University of Medical Sciences, (grant number: 1401-3-97-62306).

**Competing interests:** The authors have declared that no competing interests exist.

Fraction: 6.36(95%CI: 1.82, 10.70)] of esophageal cancer, 3% [Population Attributable Fraction: 3.43 (0, 7.33)] of gastric cancer, and 6% [Population Attributable Fraction: 6.25 (1.76, 13.63)] of colorectal cancer mortality. Using a healthy tap water source could save 5% [Population Attributable Fraction:5.50(0, 10.93)] of esophageal cancer mortality, particularly in rural areas. There was no significant association between indoor air pollution for heating purposes and animal contact with cancer mortality.

## Conclusion

Considering the results of this study, eliminating solid fuel for most daily usage, among the population with specific cancer types, is required to successfully reduce cancer related mortality. Adopting appropriate strategies and interventions by policymakers such as educating the population, allocating resources for improving the healthy environment of the community, and cancer screening policies among susceptible populations could reduce cancer related mortalities.

## Introduction

Globally, cancer is the second cause of mortality and the number of cancer patients worldwide is expected to double in the next 50 years [1, 2]. The number of disability-adjusted life years (DALYs) due to all cancer has been gradually rising over the past two decades [3]. 8.97 million deaths were caused by cancer, accounting for 15.8% of all-cause mortality [4]. The growing incidence and burden of cancers in the past decades demands thorough investigation of cancers and the epidemiology of responsible risk factors [5].

The overall impact of environmental risk factors on cancer can be measured by either the number of attributable cancer cases or the number of attributable cancer deaths. Worldwide, it is estimated that around 20% of premature cancer deaths are due to environmental risk factors. In Europe, almost 9% of all cancer deaths in 2019 were estimated to be attributable to this group of risk factors Additionally, environmental risks were responsible for more than 5% of cancer deaths in several European countries [6]. The International Agency for Research on Cancer (IARC) has identified a significant number of modifiable factors, including indoor pollutants from burning coal, as known human carcinogens [7, 8]. About 3 billion people worldwide are affected by indoor air pollution (IAP), which is mostly caused by the use of solid fuels for cooking and heating. This has a significant negative impact on health, including cancers [9]. Although generally environmental risk factors are known to be non-modifiable, the effects of some of these risks could be limited by making changes in the environment and place of living, especially regarding risks like household air pollution which could be significantly controlled by improving in-house fuel used for cooking, heating, or other purposes [10]. These statistics highlight the importance of addressing environmental and occupational risk factors for cancer through evidence-based interventions.

It is crucial to implement strategies that modify behaviors, settings, processes, regulations, and policies to reduce exposure to environmental risk factors, particularly in low middle income countries (LMIC) in which the effect of environmental risk factors on burden of disease is significant, and this major issue is neglected principally in vulnerable groups [11]. By doing so, the prevalence of these risk factors can be reduced, leading to a decrease in the number of future cancer mortality. The population attributable fraction (PAF), which is the

percentage of mortality that is attributable to that exposure, can be used to evaluate the public health impact of specific exposures on disease mortality. [12] The PAF is an appropriate epidemiologic measure to be considered by policymakers for prioritizing the preventive strategies and interventions based on the percentage of mortality saved by modifying a risk factor.

In this study, we aimed to calculate the population attributable fraction of environmental risk factors on cancer mortality in a large population-based cohort study with about 18 years of follow up which is settled in a LMIC to provide evidence for policy implication. The main findings of this study were outlining the patterns of predominant fuel use, water source, and animal contact in the cohort study and the share of mortality of each major cancer attributable to the corresponding type of each environmental risk factor.

## Materials and methods

### Study population

The Golestan Cohort Study (GCS) is a population-based cohort study that aimed to evaluate the environmental and lifestyle risk factors for cancer in Golestan Province, Iran. The study population consisted of 50,045 individuals aged 40 to 75 years, who were recruited over a period of 4 years (2004–2008). The rural participants were all eligible people living in 326 villages of GCS and were recruited through the primary health care network, while the urban participants were recruited based on random systematic clustering using household numbers. Participants who were diagnosed with cancer before enrollment, those who were unwilling or unable to proceed to the study, were excluded. All participants provided a written informed consent, and the Ethical committee of Digestive Disease Research Institute of Tehran University of Medical Sciences approved the study. More details can be found elsewhere [13]. Since enrollment, all GCS participants have been actively followed-up for endpoints by annual telephone surveys and home visits. In this study, we included all 2,196 individuals who died of cancer during the 18 years of follow-up from any type of cancer to calculate population attributable fraction of environmental risk factors.

### Data gathering

Qualified experts collected information on lifestyle, and environmental factors, including the most common fuels used for cooking and heating, current and former animal contact, and water source. The experts used valid standardized questionnaires to gather data on various demographic variables, such as age, gender, place of residence, level of education, and socio-economic status (SES). Possession of a property, house size and design, car ownership, and ownership of appliances like televisions, refrigerators, freezers, vacuum cleaners, or washing machines were used to calculate SES. The detailed method was previously described [14]. The questionnaires also assessed lifestyle risk factors such as smoking, opium use, alcohol consumption, and physical activity, as well as medical history, age of diagnosis, and medical treatments. All questionnaires were validated through a pilot study, and the details of the validation process were previously published [15].

### Assessment of the main exposure

In the Golestan Cohort Study, a variety of fuels were examined as environmental risk factors, including natural gas (liquefied petroleum gas), kerosene, and biomass fuels such as wood and animal dung. Participants were individually queried about the types of fuels they used in their homes for cooking and heating throughout their lifetime, as well as the age at which they began and, if applicable, stopped using each fuel. Exposure to indoor air pollution was defined

as the use of non-gas fuels without chimneys in households. Participants were categorized based on their predominant fuel use into three groups: "predominant biomass," "predominant kerosene," and "predominant gas." A mixed fuel category was established when no single fuel consumption accounted for more than 60% of the total assignment over a 20-year period. More detail on the categorization can be found elsewhere [8]. Animal contact in the study referred to daily interactions with ruminant animals such as cows, sheep, or goats, including activities like working with or being in proximity to these animals. Water source exposure pertained to the origin of drinking water. Participants were evaluated for their history of consuming unpiped water, typically obtained from wells or rivers. Categorization of participants was based on their exposure to drinking un-piped water, distinguishing between "never exposed" and "continued exposure" [16].

## Follow-up and outcome assessment

Each participant was followed up annually through phone calls and home visits from 2004 to November 2022. The loss to follow-up was less than one percent, indicating a high retention rate. For cancer incidence cases, the study team conducted home visits and collected relevant medical records. The diagnosis was confirmed by two independent physicians, and in cases of discrepancies, a third expert opinion was sought. To ensure accurate classification of cancers, the data from the Golestan Cohort Study was matched with the Golestan population-based cancer registry [8]. In the event of a reported death, clinical or hospital records were collected, and if necessary, a verbal examination was conducted. The 10th revision of the International Classification of Diseases and Related Health Problems (ICD-10) codes were used in the Golestan Cohort Study to classify cancer cases. Participants who died from esophageal cancer (ICD-10 Codes: C15), stomach cancer (C16), colorectal cancer (C18-C20), pancreas cancer (C25), Hematologic cancers(C81-C96), lung cancer (C34), and brain cancer (C71) were included in the study, while participants who were diagnosed with cancer at baseline (i.e. the first two years of the follow up) were excluded.

## Statistical analysis

The hazard ratio (HR) of household use of fuel, animal contact, and water source and all-cancer mortality, and specific cause cancer mortality was estimated separately using Cox proportional hazards regression models, which also provided the associated 95% confidence intervals (CIs).

The entry time was defined as the enrollment date and end of follow-up time was set at death date due to cancer. The HR of mortality associated with environmental risk factor, was adjusted for age, gender, education, SES, residence (rural/urban), smoking cigarettes, opium consumption, alcohol consumption. Due to importance of residence in rural and urban context, stratification was also implied to report the hazard and PAF in each context. More adjustment for other potential covariates such as healthy index score, energy intake, metastasis status, and indoor or outdoor job did not have any impact on the results, consequently were not included in modeling. We also excluded the first two years of follow-up to avoid potential reverse causality.

Based on the main model, the following equation was used to calculate the population attributable fraction:

$$\text{PAHF(t)} = \frac{p(HR(t) - 1)}{1 + p(HR(t) - 1)}$$

This formula corresponds to the traditional PAF formula, where Relative Risk (RR) is replaced by HR (t) obtained from survival models [17]. PUNAFCC command (the Stata package: Population Attributable and Non-attributable Fractions in Case Control or Survival Studies) was used to calculate PAF and according uncertainty intervals for survival analysis [18]. All statistical analyses were performed using Stata statistical software version 14 (Stata Corp).

## Results

### Baseline characteristics

From 50,045 participants in Golestan Cohort, 211 of them were diagnosed with cancer when entered the study, which 61 did not have complete information of their disease or of the variables we were investigating, so 49,773 participants of whom 2,196 were deceased from cancer were enrolled in this study. Of all participants, 28,685 (57.56%) were female and 21,149 (42.44%) were male. Most of them 39,866 (80%) lived in rural areas and others 9,968 (20%) lived in urban areas. The vast majority, 41,728 (84.76) used gas as their main fuel for cooking, and as for heating only 5,839 (11.86%) used gas. The main fuel for heating was Kerosene used by 35,486 (72.08) individuals. About 45 percent (22,800) of participant had former contact and 6.72% (3,346) had no animal contact in their life. 8,419 (16.91%) of participants did not have access to tap water source in their life. Gastrointestinal (GI) Cancer patients had the highest mortality with 1,099 deaths after about 18 years of follow up followed by hematologic cancer (166), lung cancer (159), and brain cancer (79). The majority of deceased cancer patients aged between 55 and 64 except for brain cancer which was a decade earlier. Most cancer deaths were reported in male participants particularly, lung and stomach cancer. This pattern is reverse in brain cancer. Mortality of cancer is higher in individuals with low score of wealth among all type of cancer except for colorectal and hematologic cancers. The prominent fuel used for cooking and heating is gas and kerosene in all cancer type, respectively. Biomass use is higher than kerosene use in stomach and esophagus cancer deaths in both cooking and heating purpose, but is in third level in other cancers. Higher amount of former animal contact is reported among stomach and colorectal cancer deaths comparing to other cancer types. More detail could be found on Table 1.

### HR and PAF of environmental risk factors on all-cancer mortality

Among indoor air pollution, biomass use in cooking purpose is associated with increased risk of all-cancer mortality with HR 1.17 (95%CI: 1, 1.38) compared to gas. However, other type of fuel for cooking and heating purpose had no association with all-cancer mortality. Of total cancer deaths, 1.26% (95% CI: 0, 2.64) were attributable to biomass use as fuel for cooking purposes.

This means that about 26 deaths from cancer could be prevented if biomass is eliminated for cooking purpose. There was no significant association between animal contact and type of water source as environmental risk factors with all-cancer mortality. There is no significant association found in rural and urban residence between exposures and mortality (Table 2).

### HR and PAF of environmental risk factors on GI cancer mortality

Predominant biomass use for cooking was associated with higher risk of esophageal, gastric and colorectal cancer mortality with HR: 1.66 (95% CI: 1.22, 2.26), 1.37 (1, 1.92), 1.84 (1, 3.42), respectively. There is no association between kerosene use and risk of GI cancer mortality. The estimated PAF for biomass was slightly higher among esophageal cancer 5.70 (1.53, 9.69) comparing to gastric cancer 3.01 (0, 6.45). Eliminating biomass for cooking purpose could reduce

**Table 1. Baseline characteristics Golestan Cohort participants and the demographic and environmental risk factors by cancer type mortality.**

| | Cohort Participants (49773) | All-cancer Mortality (2196) | Hematologic Cancer Mortality (166) | Stomach Cancer Mortality (434) | Pancreas Cancer Mortality (107) | Lung Cancer Mortality (159) | Esophagus Cancer Mortality (414) | Colorectal Cancer Mortality (144) | Brain Cancer Mortality (79) |
|---|---|---|---|---|---|---|---|---|---|
| Age | N (%) | N (%) | N (%) | N (%) | N (%) | N (%) | N (%) | N (%) | N (%) |
| 40–44 | 12966(26.02) | 235(10.70) | 20(12.05) | 33(7.60) | 7(6.54) | 20(12.58) | 32(7.73) | 15(10.42) | 13(16.46) |
| 45–54 | 20335(40.81) | 684(31.15) | 53(31.93) | 130(29.95) | 34(31.78) | 55(34.59) | 106(25.60) | 60(41.67) | 31(39.24) |
| 55–64 | 11007(22.09) | 740(33.70) | 60(36.14) | 143(32.95) | 34(31.78) | 53(33.33) | 160(38.65) | 40(27.78) | 27(34.18) |
| 65≤ | 5525(11.09) | 537(24.45) | 33(19.88) | 128(29.49) | 32(29.91) | 31(19.50) | 116(28.02) | 29(20.14) | 8(10.13) |
| Gender | | | | | | | | | |
| Female | 28658(57.56) | 949(43.21) | 74(44.58) | 131(30.18) | 50(46.73) | 42(26.42) | 185(44.69) | 67(46.53) | 46(58.23) |
| Male | 21149(42.44) | 1,247(56.79) | 92(55.42) | 303(69.82) | 57(53.27) | 117(73.58) | 229(55.31) | 77(53.47) | 33(41.77) |
| Residence | | | | | | | | | |
| Rural | 39866(80.00) | 1,799(81.92) | 129(77.71) | 382(88.02) | 81(75.70) | 121(76.10) | 379(91.55) | 101(70.14) | 60(75.95) |
| Urban | 9968(20.00) | 397 (18.08) | 37(22.29) | 52(11.98) | 26(24.30) | 38(23.90) | 35(8.45) | 43(29.86) | 19(24.05) |
| Education | | | | | | | | | |
| Illiterate | 34975(70.18) | 1,685(76.73) | 114(68.67) | 344(79.26) | 77(71.96) | 103(64.78) | 354(85.51) | 97(67.36) | 57(72.15) |
| Educated | 14859(29.82) | 511(23.27) | 52(31.33) | 90(20.74) | 30(28.04) | 56(35.22) | 60(14.49) | 47(32.64) | 22(27.85) |
| SES | | | | | | | | | |
| Q1 (Low Wealth Score) | 17841(35.80) | 941(42.85) | 53(31.93) | 190(43.78) | 42(39.25) | 59(37.11) | 222(53.62) | 49(34.03) | 35(44.30) |
| Q2 | 15415(30.93) | 688(31.33) | 61(36.75) | 144(33.18) | 28(26.17) | 48(30.19) | 130(31.40) | 41(28.47) | 26(32.91) |
| Q3 (High Wealth Score) | 16578(33.27) | 567(25.82) | 52(31.33) | 100(23.04) | 37(34.58) | 52(32.70) | 62(14.98) | 54(37.50) | 18(22.78) |
| Household Fuel For Cooking | | | | | | | | | |
| Gas | 41728 (84.76) | 1789 (82.25) | 149(89.76) | 343(79.95) | 99(92.52) | 127(81.41) | 308(74.94) | 117(81.82) | 73(93.59) |
| Kerosene | 2576(5.23) | 119 (5.47) | 6(3.61) | 21(4.90) | 1 (0.93) | 8(5.13) | 27(6.57) | 7(4.90) | 3(3.85) |
| Biomass | 2662(5.41) | 160 (7.36) | 8(4.82) | 41(9.56) | 4 (3.74) | 7(4.49) | 50(12.17) | 12(8.39) | 2(2.56) |
| Mixed nonpermanent | 2262(4.59) | 107 (4.92) | 3(1.81) | 24(5.59) | 3(2.80) | 14(8.97) | 26(6.33) | 7(4.90) | 0(0) |
| Household fuel for heating | | | | | | | | | |
| Gas | 5839(11.86) | 244 (11.22) | 34(20.48) | 27 (6.29) | 15(14.02) | 25(16.03) | 21(5.11) | 30(20.98) | 9(11.54) |
| Kerosene | 35486(72.08) | 1552(71.36) | 114(68.67) | 323 (75.29) | 72(67.29) | 98(62.82) | 306(74.45) | 88(61.54) | 59(75.64) |
| Biomass | 3389(6.88) | 195(8.97) | 9(5.42) | 47 (10.96) | 6(5.61) | 11(7.05) | 58(14.11) | 12(8.39) | 4(5.13) |
| Mixed nonpermanent | 4514(9.17) | 184(8.46) | 9(5.42) | 32 (7.46) | 14(13.08) | 22(14.10) | 26(6.33) | 13(9.09) | 6(7.69) |
| Animal Contact | | | | | | | | | |
| Never | 3346(6.72) | 130(5.92) | 14(8.43) | 17(3.92) | 9(8.41) | 11(6.92) | 23(5.56) | 5(3.47) | 7(8.86) |
| Current | 23628(47.47) | 744(33.88) | 55(33.13) | 138 (31.80) | 26(24.30) | 62(38.99) | 118(28.50) | 52(36.11) | 37(46.84) |
| Former | 22800(45.81) | 1322(60.20) | 97(58.43) | 279(64.29) | 72(67.29) | 86(54.09) | 273(65.94) | 87(60.42) | 35(44.30) |
| Water Source | | | | | | | | | |
| Tap Water | 41354(83.09) | 1798(81.88) | 137(82.53) | 354(81.57) | 96(89.72) | 134(84.28) | 313(75.60) | 124(86.11) | 62(78.48) |
| Other source | 8419(16.91) | 398(18.12) | 29(17.47) | 80(18.43) | 11(10.28) | 25(15.72) | 101(24.40) | 20(13.89) | 17(21.52) |

about 23 and 13 deaths from esophageal and gastric cancer. Although biomass was associated with higher risk of colorectal cancer mortality, but the respective PAF was not significant. Moreover, no association was found between different type of fuel use as indoor pollution for heating propose and GI cancer mortality. Using non-tap water source is associated with higher risk of esophageal cancer mortality with HR 1.26 (1.00, 1.58), and PAF: 5.01 (0, 10.03), saving about 21 deaths by using healthy tap water source. Hence, there is no association reported

**Table 2. The hazard rate and population attributable fraction of environmental risk factors on all-cancer mortality.**

| | Total | | Rural | | Urban | |
|---|---|---|---|---|---|---|
| | HR* (95% CI) | PAF% (95% CI) | HR^ | PAF | HR^ | PAF |
| Indoor Pollution(Recent Cooking Predominant fuel) | | | | | | |
| Gas | 1 | | 1 | | 1 | |
| Biomass | **1.17(1, 1.38)** | **1.26(0, 2.64)** | 1.17(0.98,1.38) | 1.52(-0.18,3.21) | 0.59(0.08,4.25) | NA |
| Kerosene | 1.01(0.83, 1.21) | 0.06(-1.17,1.29) | 1.06(0.87,1.29) | 0.53(-0.96, 2.00) | 0.65(0.33,1.27) | NA |
| Mixed non-prominent | 1.04(0.86,1.27) | 0.26(-0.86,1.37) | 1.09(0.88,1.34) | 0.55(-0.78,1.87) | 0.76(0.42,1.36) | NA |
| Indoor Pollution (Recent Heating Predominant fuel) | | | | | | |
| Gas | 1 | | 1 | | 1 | |
| Biomass | 1.00 (0.78, 1.28) | NA | 0.80(0.54,1.18) | NA | 0.38(0.05,2.78) | NA |
| Kerosene | 0.88(0.72,1.08) | NA | 0.71(0.49,1.02) | NA | 1.05(0.79,1.40) | 1.21(-5.59,7.59) |
| Mixed non-prominent | 0.90(0.74,1.10) | NA | 0.72(0.47,1.11) | NA | 0.93(0.74,1.17) | NA |
| Former Animal Contact | | | | | | |
| Yes | 0.96(0.78, 1.14) | NA | 089(0.73,1.08) | NA | 1.43(0.87,2.35) | 29.78(-12.01,55.98) |
| Water Source | | | | | | |
| Tap water | 1 | | 1 | | 1 | |
| Other | 1.01(0.91,1.13) | 0.30(-1.70, 2.28) | 1.02(0.91,1.14) | 0.54(-1.90,2.92) | 0.35(0.05,2.54) | NA |

*Adjusted for age, gender, education, SES, residence (rural/urban), smoking cigarettes, opium consumption, and alcohol consumption

^Adjusted for age, gender, education, SES, smoking cigarettes, opium consumption, and alcohol consumption

with other type of GI cancer mortality. There is no association between animal contact and GI cancer mortality (Table 3).

In rural area, biomass use for cooking propose was associated with higher risk of esophageal cancer mortality [HR: 1.69(1.24, 2.30), PAF: 6.36(1.82, 10.70)], gastric cancer mortality [HR:

**Table 3. The hazard rate and population attributable fraction of environmental risk factors on GI-cancer mortality.**

| | Esophageal Cancer | | Gastric Cancer | | Colorectal Cancer | |
|---|---|---|---|---|---|---|
| | HR* (95% CI) | PAF% (95% CI) | HR^ | PAF | HR^ | PAF |
| Indoor Pollution(Recent Cooking Predominant fuel) | | | | | | |
| Gas | 1 | | 1 | | 1 | |
| Biomass | **1.66 (1.22, 2.26)** | **5.70 (1.53, 9.69)** | **1.37(1, 1.92)** | **3.01 (0, 6.45)** | **1.84(1, 3.42)** | 4.40 (-1.31, 9.80) |
| Kerosene | 1.21(0.81, 1.79) | 1.51 (-1.92, 4.83) | 0.90(0.58, 1.41) | NA | 1.14(0.53, 2.46) | 0.79 (-4.10, 5.45) |
| Mixed non-prominent | 1.29(0.86, 1.93) | 1.77 (-1.33, 4.79) | 1.15(0.76, 1.75) | 0.90 (-1.83, 3.56) | 1.18(0.54, 2.54) | 0.86 (-3.53, 5.07) |
| Indoor Pollution (Recent Heating Predominant fuel) | | | | | | |
| Gas | 1 | | 1 | | 1 | |
| Biomass | 1 | | 1.65(0.89, 3.05) | 25.65 (-7.47, 48.57) | 0.91(0.36, 2.27) | NA |
| Kerosene | 1.62 (0.85, 3.08) | 28.77 (-12.12, 54.75) | 1.33(0.77, 2.29) | 23.19 (-26.19, 53.25) | 0.67(0.33, 1.36) | NA |
| Mixed non-prominent | 1.09 (0.60, 1.96) | 08.15 (-59.26, 47.03) | 1.29(0.76, 2.20) | 12.29 (-16.61, 34.03) | 0.60 (0.31, 1.17) | NA |
| Former Animal Contact | | | | | | |
| Yes | 1.01 (.65, 1.56) | NA | 1.42 (0.86, 2.33) | 28.08 (-14.95, 55.00) | 1.90 (0.76, 4.74) | 43.33 (-34.64, 76.15) |
| Water Source | | | | | | |
| Tap water | 1 | | 1 | | 1 | |
| Other | **1.26 (1.00, 1.58)** | **5.01 (0, 10.03)** | 0.93(0.72, 1.19) | NA | 0.88(0.54, 1.45) | NA |

*Adjusted for age, gender, education, SES, residence (rural/urban), smoking cigarettes, opium consumption, and alcohol consumption

^Adjusted for age, gender, education, SES, smoking cigarettes, opium consumption, and alcohol consumption

1.38 (1, 1.93), PAF: 3.43 (0, 7.33)], and colorectal cancer [HR: 1.88(1.00, 3.53), PAF: 6.25 (1.76, 13.63)]. This pattern is not seen in urban area. Moreover, non-tap water source is associated with higher risk of mortality in esophageal cancer [HR: 1.26(1.01,1.59), PAF:5.50(0, 10.93)]. There was no association in urban area between environmental risk factors and cancer mortality. In addition, no association was found between environmental risk factors and pancreatic cancer mortality.

### HR and PAF of environmental risk factors on non-GI cancer mortality

Among included non-GI cancers, hematologic cancers, lung cancer, and brain cancer had the highest mortality number, respectively. However, the mentioned cancers' mortality was low comparing to GI mortality. Consequently, the estimated hazard and PAF for environmental risk factors were prone to wide uncertainties. Among, included environmental risk factors, biomass use for cooking in urban area with HR:7.29(1, 53.92) was associated with higher risk of hematologic cancer mortality but the associated PAF was non-significant due to low prevalence of exposure in the study. Moreover, mixed fuel use for cooking is associated with higher risk of lung cancer mortality with HR: 1.92(1.10,3.36), PAF: 4.76%(0, 9.85). The HR [2.85(1, 8.13)] was also significant in urban area but not the PAF [6.88(-3.99,16.62)].

## Discussion

This study estimated the population attributable fraction of cancers due to environmental risk factors. The main finding of this study was a significant reduction in cancer-related mortality when biomass fuel, as an indoor air pollution, was replaced with gas for cooking purpose, particularly in GI cancers. Prominent use of biomass for cooking purpose increased 84%, 66%, and 37% the risk of colorectal, esophageal, and gastric cancer mortality, respectively. About 6 percent of esophageal cancer, 3% of gastric cancer, 6% of colorectal cancer mortality would be saved by eliminating non-gas fuel for cooking particularly in rural area. Using healthy piped water source could save 5% of esophageal cancer mortality, particularly in rural area. We could not find any significant association between indoor air pollution for heating purpose or animal contact with cancer mortality. Although, accessing sufficient healthcare facilities and quality treatment is a challenge for those living in rural areas, controlling environmental risk factors could potentially reduce mortality in both rural and urban regions [19].

Indoor air pollution from solid fuels is a major risk of respiratory tract and other organs diseases and cancers, especially when solid fuels are used in poorly ventilated spaces in which toxic emissions from combustion accumulate and increase the risk of diseases [20]. The aforementioned situation of using solid fuels in poorly ventilated places mainly happens in developing and low- and middle-income countries, where poor housing structures exist and solid fuels are still extensively used to heat the living place and for cooking purposes [21]. The contribution of indoor air pollution from solid fuels to lung cancer and other respiratory cancers like those arising from larynx, hypopharynx, tracheal, and bronchus is demonstrated previously [22]. Moreover, in low-income population, biomass smoke is a risk factor for esophageal and gastric cancer [8]. A meta-analysis of literature found higher risk of upper-digestive tract and cervical cancer associated with household air pollution, while the results were adjusted for other risks like smoking [23]. Studies have shown associations between certain types of air pollution and brain cancer [24, 25]. Besides, some studies have reported associations between indoor air pollution from solid fuels and other neurologic conditions and cognitive impairments, which altogether suggests the neurologically toxic impact of the emissions of solid fuels on human brain and neurologic system [26–28]. In this study, we did not observe any impact of indoor air pollution on the mortality rate of brain cancer patients. Among hematologic

malignancies, studies reported associations between indoor pollutants of nitrogen dioxide and several types of volatile organic compounds and incidence of childhood acute leukemia [26, 29]. It is suggested that various environmental pollutants added to immunologic factors and genetic susceptibility may expose to leukemia [30, 31]. Parallel to other studies, we observed that the use of solid fuels for cooking purpose had a significant impact on esophageal, gastric, colorectal, lung, and hematologic cancer related mortality particularly in individuals living in rural areas. By implementing appropriate interventions to minimize the application of solid fuel by society, 1% of all-cancer, 6% of esophageal, 3% of gastric, and 6% of colorectal cancer deaths would be averted, adjusted for confounders including socioeconomic status. This has great impact on the burden of cancers in LMIC which are struggling with the health inequities.

According to several studies, there is an apparent excess risk of all-cancer mortality among cities that use surface drinking water supplies, with the increased risk being slight but statistically significant, while the results of recent studies do not appear to implicate drinking water as a factor in causing considerably higher cancer mortality rates in cities, and the gastric cancer mortality rate is higher among those who drink river water than those who drink shallow well water [32–34]. A study by the Environmental Working Group found that millions of Americans are drinking water containing carcinogens, and the contamination may be responsible for more than 100,000 cases of cancer. The strongest evidence for a cancer risk involves arsenic, which is linked to cancers of the liver, lung, bladder, and kidney [35, 36]. In LMIC, the relationship between water source and cancer mortality is not extensively studied. This indicates a gap in research in these regions. However, the available previous evidence suggests that water source can potentially impact cancer mortality rates. This study implemented in a LMIC for the first time suggested association between water source and esophageal cancer, which is parallel to available evidence from developed countries.

Epidemiologic studies to date have provided little evidence that animal viruses and bacteria cause human cancer, although exposure to farm animals or manure has been associated with childhood brain and adult hematologic tumors in some studies [37, 38]. Overall, the available search results do not provide clear evidence on the relationship between animal contact and cancer mortality in humans. This study did not find any association between animal contact and any type of cancer mortality.

Iran, as a LMIC, faced a dramatic change in the burden of cancer and exposure to environmental risk factors in recent years. Exposures like indoor air pollution due to solid fuels have decreased substantially with wide-spread improvements in gas piping system through the country in this period [39, 40]. However, the use of biomass and solid fuels still is the only option in some regions of the country with limited resources [40]. Public health implications of the findings of this study could be focused on the populations with significant exposure to indoor air pollutants due to solid fuels and provide more cancer screening programs for them. Also, continuing efforts to decrease solid fuel use for heating, and cooking especially in settings with limited and inadequate ventilation need to be pursued to handle the burden of cancers associated with the toxic emissions of these sources of energy. To reduce the burden of cancer, it is crucial to implement evidence-based and comprehensive cancer prevention strategies that target modifiable risk factors. These strategies can include public policies, health-promoting environments, personal and community interventions, and optimized health service. By implementing interventions that target these risk factors, it is possible to make significant progress in reducing the number of cancer mortalities [41]. Reducing the environmental risk factors that affect cancer mortality requires a multi-faceted approach [42]. First, allocating sufficient resources by regulations to prevent exposure (including occupational) and reducing pollution. Stakeholders should allocate resources to mitigate or eliminate barriers to reducing environmental cancer risks. Second, changing behaviors can also reduce environmental and

occupational cancer risk. Policies should be implemented to provide education and awareness to the public about the environmental risk factors that can cause cancer. This can include information about how to reduce exposure to indoor air pollution and how to identify hazardous substances. Third, providing access to clean water should be implemented to ensure that all populations have access to clean water sources. Fourth, conducting research on the environmental risk factors that can cause cancer. This can include identifying the sources, magnitude, routes of exposure, and distribution of these risk factors. Measuring the potential cancer mortality related to environmental hazard exposure can benefit to update efforts to modify health of communities. Finally, environmental justices are necessary and one of the social determinants of health which is neglected particularly in LMIC. The underprivileged population should be priories in implementation of policies which reduce the unique or compounded health risks. By implementing these policy strategies, it is possible to minimize environmental risk factors that affect cancer mortality and reduce the burden of cancer. However, it is important to prioritize efforts and focus on equity when implementing cancer prevention strategies. This ensures that interventions reach all populations, including vulnerable and marginalized groups, and that resources are allocated appropriately. Timely implementation, focus on equity, and prioritization of efforts are associated with public health and economic benefits. Detecting the level of awareness of cancer risk among high risk groups is essential for achieving health equity. Providing access to screening for people who are medically underserved and have low incomes, studying the cost of preventive care and cancer treatment, and the reasons people may not be able to prevent cancer, early detection, and treatment are other ways to achieve equity.

The strength of this study is its extensive follow-up of a large population based cohort with detailed risk factor assessments, and low rates of loss to follow-up which enable us to calculate precise estimations. Furthermore, this study is among the first to assessing the proportion of cancer mortality attributed to environmental risk factors in a LMIC. However, we could not assess the quantitative markers of indoor air pollution, such as the amount of natural carcinogenic chemicals produced by burning fuel in the household air as the data were not available. In this study, the adjusted Hazard Ratio (HR) was used to estimate Population Attributable Fraction (PAF). however, other variables such as access to facilities, first diagnosis, and treatment may have an impact on the prevalence and hazard ratios. we used socioeconomic status and urbanization as a proxy for health care access as they have correlation, hence, future research should investigate these factors in further studies. Moreover, there are other sources of indoor pollution that were not assessed in this cohort study. Bakhoor and other types of incense pose adverse health effects when used indoors, particularly in locations with inadequate ventilation. These suggested to be considered in future cohort studies.

## Conclusions

To conclude, the effect of environmental risk factors is significant with substantial number of cancer mortality attributable to these risk factors. This study found significant associations between certain types of cancers and exposure to indoor air pollution and unsafe water source, based on a population-based cohort study in Iran. GI cancer showed to have strong associations with fuel and water source, especially among rural residents, which are more exposed to solid fuels use in Iran. Adopting appropriate strategies and interventions such as educating population, allocating resources for improving healthy environment of the community, and cancer screening polices among susceptible populations and decreasing exposure to solid fuels is suggested to be adopted by stakeholders and policymaker to reduce cancer mortality.

## Author Contributions

**Conceptualization:** Negar Rezaei, Maryam Sharafkhah, Sadaf G. Sepanlou.

**Formal analysis:** Maryam Sharafkhah, Sahar Dalvand, Sahar Masoudi.

**Methodology:** Negar Rezaei, Sadaf G. Sepanlou, Hossein Poustchi, Alireza Sajadi.

**Project administration:** Hossein Poustchi, Gholamreza Roshandel, Masoud Khoshnia, Layli Eslami, Alireza Delavari.

**Supervision:** Alireza Delavari.

**Writing – original draft:** Negar Rezaei, Yalda Farahmand.

**Writing – review & editing:** Negar Rezaei, Maryam Sharafkhah, Yalda Farahmand, Sadaf G. Sepanlou, Sahar Dalvand, Hossein Poustchi, Alireza Sajadi, Sahar Masoudi, Gholamreza Roshandel, Masoud Khoshnia, Layli Eslami, Mahboube Akhlaghi, Alireza Delavari.

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
