## [Decision Letter · Decision Letter 0]

13 Mar 2024

PONE-D-23-36813Population Attributable Fractions of Cancer Mortality Related to Environmental Risk Factors: The Golestan Cohort StudyPLOS ONE

Dear Dr. Delavari,

Thank you for submitting your manuscript to PLOS ONE. After careful consideration, we feel that it has merit but does not fully meet PLOS ONE’s publication criteria as it currently stands. Therefore, we invite you to submit a revised version of the manuscript that addresses the points raised during the review process.

As an Academic editor I would like you to, besides the reviewers' comments, pay attention to few more things as well: I strongly recommend you to read carefully reviewers' comments and try to answer them as much as possible to correlate with their opinions and requirements. Please do all the necessary changes.

I require the explanation about data availability - in your submission this is not well explained nor in accordance with journal's propositions. Please check and revise it.

I would like you to add the conclusion part in abstract instead of discussion.

Overall, the paper has potential but it needs to be carefully arranged to correspond to the title - maybe to consider the word "specific" in front of environmental risk factors while you did not check all the possible environmental risk factors...

We look forward to receiving your revised manuscript.

Kind regards,

Iskra Alexandra Nola

Academic Editor

PLOS ONE

Journal Requirements:

"The Golestan Cohort Study work was funded by the Tehran University of Medical Sciences (grant number: 81/15), Cancer Research UK (grant number: C20/A5860), the Intramural Research Program of the U.S. National Cancer Institute, National Institutes of Health, and the International Agency for Research on Cancer. The presented analysis was supported by Digestive Disease Research Institute, Tehran University of Medical Sciences, (grant number: 1401-3-97-62306)"

Reviewers' comments:

Reviewer's Responses to Questions

**Comments to the Author**

1. Is the manuscript technically sound, and do the data support the conclusions?

Reviewer #1: Yes

Reviewer #2: Yes

2. Has the statistical analysis been performed appropriately and rigorously? 

Reviewer #1: Yes

Reviewer #2: Yes

3. Have the authors made all data underlying the findings in their manuscript fully available?

Reviewer #1: Yes

Reviewer #2: Yes

4. Is the manuscript presented in an intelligible fashion and written in standard English?

Reviewer #1: Yes

Reviewer #2: Yes

5. Review Comments to the Author

Reviewer #1: Population Attributable Fractions of Cancer Mortality Related to Environmental Risk Factors: The Golestan Cohort Study

This article entitled: Population Attributable Fractions of Cancer Mortality Related to Environmental Risk Factors: The Golestan Cohort Study, seems an interesting topic that provides another face of environmental risk factors related to cancer mortality.

Some comments may need to be reviewed and answered.

1- The abstract showed no conclusion as a subheading.

2- On page 7, within the methodology section under "Follow-Up and Outcome Assessment," at line 138: It's essential to specify the time frame for follow-up in this section, indicating when it commenced and over how many years it extended. Furthermore, clarification is needed regarding whether participants encompassed both cancer and non-cancer patients.

3- On page 9, within the Statistical Analysis section, at line 169: Abbreviations like RR and PUNAFCC were not sufficiently defined for clarity.

4- On page 10, in the results section, at line 188: Is the elevated cancer mortality rate among individuals with low wealth attributed to limited access to specialized cancer treatment centers in rural areas, leading to inadequate monitoring of patient status compared to those residing in urban areas within the studied demographic?

Should late diagnosis be considered a factor more significant than environmental factors?

5- The primary findings of this study necessitate adjustments to clinical services to guarantee timely access to sufficient healthcare facilities and top-notch diagnosis and treatment. It's important to note that individuals in rural areas frequently encounter difficulties in accessing such services.

6- On page 16, within the discussion section, at line 264: The indicated 6% may require comparison with the figures presented in the selected table.

7- On page 16, in the discussion section, at line 284: The limited influence of indoor pollution on brain cancer, as observed in this study, may be associated with the metastatic progression of the disease, particularly evident in advanced-stage cases.

8- On page 17, in the discussion section, at line 289: Consistent with prior research, our findings indicate that the utilization of solid fuels for cooking is notably associated with higher mortality rates linked to esophageal, gastric, colorectal, lung, and hematologic cancers, especially among individuals residing in rural regions. This observation suggests that those who rely on solid fuels for cooking are often of lower socioeconomic status and likely face challenges accessing adequate healthcare facilities for optimal diagnosis and follow-up. Therefore, these findings appear to be more closely linked to the accessibility of healthcare services rather than solely to environmental factors.

9-

Page 20, line 357: why the author did not mentioned the role of facility use as indicator to delay mortality as well as lead to early diagnosis to avoid this type of confounder on the given findings.

Alternatively, could the author elaborate on whether Bahkoor and other types of incense pose adverse health effects when used indoors, particularly in locations with inadequate ventilation?

Reviewer #2: Title:

Population Attributable Fractions of Cancer Mortality Related to Environmental Risk Factors: The Golestan Cohort Study

Since cancer is the major health challenge globally, i would like to thank the authors for drafting this manuscript. The manuscript is well organized and scientifically sounds for publication. Please find my concerns below.

Concern one on title: The phrase “environmental factors” are more general as compared to environmental risk factors attributable to cancer mortality included in this study. It is very difficult to include all environmental risk factors attributable to cancer mortality in this study. Hence, this needs to be justified and/or modification of the title.

Concern two: Abstract discussion Section: It is not recommended to add discussion in the abstract section. Instead it is highly advisable to include conclusion and highlight of the recommendation section about what to do to reduce cancer mortality as a result of exposure to various environmental risk factors.

Concern Three: introduction section: Line 59-60: “Worldwide, it is estimated 60 that around 20% of premature cancer deaths are due to environmental risk factors” If this is known, what is the unknown here? Please provide your justification.

Concern: four Introduction section: Is there similar articles published before? Or is it the only study on the issue? If there is previous similar work, what was the gap the authors identified and wants to fill, so that the international community can gain new insights?

Concern Five: How cofounders can practically be controlled. A cancer patient may be died during follow up period because of other co-morbidities.

6. PLOS authors have the option to publish the peer review history of their article (what does this mean?). If published, this will include your full peer review and any attached files.

Reviewer #1: **Yes: **Amen Bawazir

Reviewer #2: No

---

## [Author Response · Author response to Decision Letter 0]

27 Apr 2024

Dear Editor, 

Thank you for giving us the opportunity to submit a revised draft of our manuscript. We also thank the reviewers for their constructive and positive comments and suggestions. The changes for this revision are done, and we also provided the response letter for this revision. Two versions of the revised manuscript are provided: one clean copy and one that shows the changes indicated by the "Track Changes" function. Thanks again to you and all reviewers for your consideration and time.

As per the editor's requisition, we removed the funding information from the acknowledgment section. Also, we have presented the grant information below to change the funding statement on the online submission form on our behalf.

“The Golestan Cohort Study work was funded by the Tehran University of Medical Sciences (grant number: 81/15), Cancer Research UK (grant number: C20/A5860), the Intramural Research Program of the U.S. National Cancer Institute, National Institutes of Health, and the International Agency for Research on Cancer. The presented analysis was supported by Digestive Disease Research Institute, Tehran University of Medical Sciences, (grant number: 1401-3-97-62306).”

Sincerely,

Alireza Delavari

Here is a response to the reviewers’ comments and concerns:

Reviewer 1: 

Comment 1: The abstract showed no conclusion as a subheading.

Response: Thanks for the comment, we revised this based on your comments. 

Comment 2: On page 7, within the methodology section under "Follow-Up and Outcome Assessment," at line 138: It's essential to specify the time frame for follow-up in this section, indicating when it commenced and over how many years it extended. Furthermore, clarification is needed regarding whether participants encompassed both cancer and non-cancer patients. 

Response: Thank you for the comment. We have made the necessary modifications to this section as per your comment. Additionally, we have stated in the text that any participants who were diagnosed with cancer within the first two years of the study were excluded from the sample. Therefore, all the participants included in the first stage of the study were non-cancer participants.

Comment 3: On page 9, within the Statistical Analysis section, at line 169: Abbreviations like RR and PUNAFCC were not sufficiently defined for clarity.

Response:

We have included the complete definitions of these abbreviations.

Comment 4: On page 10, in the results section, at line 188: Is the elevated cancer mortality rate among individuals with low wealth attributed to limited access to specialized cancer treatment centers in rural areas, leading to inadequate monitoring of patient status compared to those

residing in urban areas within the studied demographic?

Should late diagnosis be considered a factor more significant than environmental factors?

Response: It is an insightful comment. It is the reason that in this study the HR and PAF for Rural and Urban areas were calculated separately. Also, HR is calculated by adjusting other confounders including SES (Social Economic Status) and urbanization as discussed in the method section to minimize this confounding effect of the factors and evaluating HR of environmental factors adjusted for mentioned confounders. In addition, the metastasis was included as a potential confounder in modeling and did not change the results. This is added in the method section. 

Comment 5: The primary findings of this study necessitate adjustments to clinical services to guarantee timely access to sufficient healthcare facilities and top-notch diagnosis and treatment. It's important to note that individuals in rural areas frequently encounter difficulties in accessing such services.

Response: The authors agree. Based on the aim of the study, we conclude that although access to services may affect the mortality rate in rural regions, controlling environmental risk factors could potentially reduce mortality in both rural and urban regions. Also, we added the mentioned note in the discussion section. Also, we cited a paper as a reference.

Comment 6: On page 16, within the discussion section, at line 264: The indicated 6% may require comparison with the figures presented in the selected table.

Response: We have commented and highlighted the related reports in the result section. This is the PAF of each cancer in rural areas and there is no figure to compare. 

Comment 7: On page 16, in the discussion section, at line 284: The limited influence of indoor pollution on brain cancer, as observed in this study, may be associated with the metastatic progression of the disease, particularly evident in advanced-stage cases.

Response:

Based on the results, the authors didn't observe the impact of indoor pollution on brain cancer patient mortality rates. In addition, the adjusted HR is calculated for the metastasis variable and we added this in the method section. The results were not changed. So, we didn’t include them in the final model. 

Comment 8: On page 17, in the discussion section, at line 289: Consistent with prior research, our findings indicate that the utilization of solid fuels for cooking is notably associated with higher mortality rates linked to esophageal, gastric, colorectal, lung, and hematologic cancers, especially among individuals residing in rural regions. This observation suggests that those who rely on solid fuels for cooking are often of lower socioeconomic status and likely face challenges accessing adequate healthcare facilities for optimal diagnosis and follow-up. Therefore, these findings appear to be more closely linked to the accessibility of healthcare services rather than solely to environmental factors.

Response: Thanks for the insightful comment. As the reviewer correctly indicated there is an association between SES and healthcare access which is why we used SES as an indicator in our modeling to adjust the confounding effect of this on the effect of environmental factors on mortality. In the modeling, all available confounder variables were added to the model to minimize the effect of other confounder factors like SES on the final HR of environmental factors. The effect reported in this study is adjusted for not only SES, but stratified for Rural and Urban which both correlate with healthcare access and all the according benefits like early diagnosis and treatment. Therefore, the effect reported for environmental factors addresses all above mentioned confounding effects. In this part of the discussion, we just describe the associations in low-middle-income countries. We have added more clarification on this in the discussion section. 

Comment 9: Page 20, line 357: why the author did not mention the role of facility use as an indicator to delay mortality as well as lead to early diagnosis to avoid this type of confounder on the given findings?

Response:

Thank you for your comment. As the reviewer correctly indicated there is an association between SES and healthcare access and use which is why we used SES as an indicator in our modeling to adjust the confounding effect of this on the effect of environmental factors on mortality. In the modeling, all available confounder variables were added to the model to minimize the effect of other confounder factors like SES on the final HR of environmental factors. The effect reported in this study is adjusted for not only SES, but stratified for Rural and Urban residents which both correlate with health care access and use, and all the benefits like early diagnosis and treatment. Therefore, the effect reported for environmental factors addresses all above mentioned confounding effects.

Comment 9: 

Alternatively, could the author elaborate on whether Bahkoor and other types of incense pose adverse health effects when used indoors, particularly in locations with inadequate ventilation?

The questionnaire didn’t contain any information about the Bahkoor and it is needed to pay attention to it as another source of indoor air pollution.

Response: Thanks for your comment. In the Golestan Cohort study questionnaire, this variable was not measured. As the reviewer correctly indicated this is an important variable and we added this to our limitation and suggested the need for future research on this in discussion. 

Reviewer 2: 

Comment 1: The phrase “environmental factors” is more general as compared to environmental risk factors attributable to cancer mortality included in this study. It is very difficult to include all environmental risk factors attributable to cancer mortality in this study. Hence, this needs to be justified and/or modification of the title.

Response: With many thanks for this clarifying comment, we changed the title as follows:

“Population Attributable Fractions of Cancer Mortality Related to Indoor Air Pollution, Animal Contact, and Water Source as Environmental Risk Factors: Findings from The Golestan Cohort Study”

Comment 2: Abstract Discussion Section: It is not recommended to add discussion in the abstract section. Instead, it is highly advisable to include a conclusion and highlight the recommendation section about what to do to reduce cancer mortality as a result of exposure to various environmental risk factors. 

Response: Thanks for the comment, we revised the abstract by removing the discussion and adding the conclusion.

Comment 3: introduction section: Line 59-60: “Worldwide, it is estimated 60 that around 20% of premature cancer deaths are due to environmental risk factors” If this is known, what is the unknown here? Please provide your justification. 

Response: Thanks for the valuable comment which highlights the most important aim of this study, as we have indicated in the introduction, It is crucial to implement strategies that modify behaviors, settings, processes, regulations, and policies to reduce exposure to environmental risk factors, particularly in low middle-income countries (LMIC) in which the effect of environmental risk factors on burden of disease is significant, and this major issue is neglected principally in vulnerable groups. By doing so, the prevalence of these risk factors can be reduced, leading to a decrease in the number of future cancer mortality. The population attributable fraction (PAF), which is the percentage of mortality that is attributable to that exposure in the setting of each country due to different prevalence, can be used to evaluate the public health impact of specific exposures on disease mortality. The PAF is an appropriate epidemiologic measure to be considered by policymakers for prioritizing the preventive strategies and interventions based on the percentage of mortality saved by modifying a risk factor and must be reported based on region prevalence of exposure which highlights the need to have reports in LMIC which is neglected. 

In this study, we aimed to calculate the population-attributable fraction of environmental risk factors on cancer mortality in a large population-based cohort study with about 18 years of follow up which is settled in a LMIC to provide evidence for policy implication. This information is provided in the introduction. 

Comment 4: four Introduction section: Are there similar articles published before? Or is it the only study on the issue? If there is previous similar work, what was the gap the authors identified and want to fill so that the international community can gain new insights? 

Response: Thank you for your insightful comment. While there have been limited studies examining the association of environmental risk factors with cancer mortality, our research is unique in its scope and focus to assess the PAF of environmental risk factors on cancer mortality from a cohort with more than 20 years follow up from a LMIC which is needed for policy implication.

Our study is distinctive in that it calculates the population-attributable fraction of environmental risk factors on cancer mortality within a large population-based cohort study over an extended follow-up period of approximately 18 years. This study is set in a low-to-middle-income country (LMIC), providing a unique perspective and valuable data for policy implications.

The primary findings of our study outline the patterns of predominant fuel use, water source, and animal contact within the cohort study. Furthermore, we have quantified the share of mortality for each major cancer type attributable to the corresponding environmental risk factor.

This approach allows us to identify specific environmental risk factors contributing to cancer mortality in this context, which can guide targeted interventions and policies. Therefore, our study fills a critical gap in the literature by providing comprehensive, long-term data on the population-attributable fraction of environmental risk factors on cancer mortality in an LMIC setting.

Comment 5: How cofounders can practically be controlled. A cancer patient may die during the follow-up period because of other co-morbidities. 

Response: Thanks for your comment, in this study, we utilized the Hazard Ratio in the PAF formula, which was estimated using a Cox regression model. This modeling method allowed us to adjust for the effect of confounders. The HR of mortality associated with environmental risk factors was adjusted for age, gender, education, SES, residence (rural/urban), smoking cigarettes, opium consumption, and alcohol consumption. Due to the importance of residence in rural and urban contexts, stratification was also implied to report the hazard and PAF in each context. More adjustments for other potential covariates such as healthy index score, energy intake, and indoor or outdoor job did not have any impact on the results and, consequently were not included in modeling. We also excluded the first two years of follow-up to avoid potential reverse causality. It is important to note that we have included all patients who have died from cancer based on ICD10. In the event of a reported death, clinical or hospital records were collected, and if necessary, a verbal examination was conducted. The 10th revision of the International Classification of Diseases and Related Health Problems (ICD-10) codes was used in the Golestan Cohort Study to classify cancer cases. Participants who died from esophageal cancer (ICD-10 Codes: C15), stomach cancer (C16), colorectal cancer (C18-C20), pancreas cancer (C25), Hematologic cancers(C81-C96), lung cancer (C34), and brain cancer (C71) were included in the study, while participants who were diagnosed with cancer at baseline were excluded. We aim to assess The hazard ratio (HR) of household use of fuel, animal contact, and water source and all-cancer mortality, and specific cause cancer mortality using Cox proportional hazards regression models, which also provided the associated 95% confidence intervals (CIs). Then we used this to estimate PAF. In this case, competitive outcome survival analysis is not applicable due to aim of the this study.

---

## [Editor Report · Decision Letter 1]

16 May 2024

Population Attributable Fractions of Cancer Mortality Related to Indoor Air Pollution, Animal Contact, and Water Source as Environmental Risk Factors: Findings from The Golestan Cohort Study

PONE-D-23-36813R1

Dear Dr. Alireza Delavari,

We’re pleased to inform you that your manuscript has been judged scientifically suitable for publication and will be formally accepted for publication once it meets all outstanding technical requirements.

Kind regards,

Iskra Alexandra Nola

Academic Editor

PLOS ONE

Additional Editor Comments (optional):

Dear dr Delavari,

I would like you to check once again the typos in your revised version (e.g. line 373).

Thank you,

Kind regards,

Iskra A. Nola
---

## [Editor Report · Acceptance letter]

30 May 2024

PONE-D-23-36813R1 

PLOS ONE

Dear Dr. Delavari, 

I'm pleased to inform you that your manuscript has been deemed suitable for publication in PLOS ONE. Congratulations! Your manuscript is now being handed over to our production team.

Kind regards, 

on behalf of

Dr. Iskra Alexandra Nola 

Academic Editor

PLOS ONE